



**A foehn-induced haze front in Beijing: observations and implications**
Ju Li[1*], Zhaobin Sun[1], Donald H. Lenschow[2], Mingyu Zhou[3], Youjun Dou[1], Zhigang Cheng[1],
Yaoting Wang[1], Qingchun Li[1]
[1] Institute of Urban Meteorology, Beijing,China
[2] National Center for Atmospheric Research, Boulder, CO, USA
[3] National Marine Environment Forecast Center, Beijing, China
*Correspondence to*: Ju Li (jli@ium.cn)
**Abstract**. Despite frequent foehns in the Beijing-Tianjin-Hebei (BTH) region, there are only a few
studies of their effects on air pollution in this region, or elsewhere. Here, we discuss a
foehn-induced haze front (HF) event using observational data to document its structure and
evolution. Using a dense network of comprehensive measurements in the BTH region, our
analyses indicate that the foehn played an important role in the formation of the HF with
significant impacts on air pollution. Northerly warm-dry foehn winds, with low particulate
concentration in the northern area, collided with a cold-wet polluted air mass to the south and
formed an HF in the urban area. The HF, which is associated with a surface wind convergence line
and distinct contrasts of temperatures, humidity and pollutant concentrations, resulted in an
explosive growth of particulate concentration. As the plains-mountain wind circulation was
overpowered by the foehn, a weak pressure gradient due to the different air densities between air
masses was the main factor forcing advances of the polluted air mass into the clean air mass,
resulting in severe air pollution over the main urban areas. Our results show that the foehn can
affect air pollution through two effects: direct wind transport of air pollutants, and altering the air
mass properties to inhibit boundary-layer growth and thus indirectly aggravating air pollution.
This study highlights the need to further investigate the foehn and its impacts on air pollution in
the BTH region.



## 1 Introduction

Air pollution issues in China have been widely discussed and studied in recent decades. The region encompassing Beijing City, Tianjin City, and Hebei province, i.e. the Beijing-Tianjin-Hebei (BTH) region, is one of most polluted areas in China and has a very high level of particular matter of 2.5 μm diameter ($PM_{2.5}$) (Han et al., 2014; Gao et al., 2015; Jiang et al., 2015a; Wang et al., 2015). Furthermore, severe haze events occur frequently in the BTH region, especially in autumn and winter (Wang et al., 2013; Sun et al., 2014; Sun et al., 2016; Zhang et al., 2015; Li et al., 2016; Li et al., 2017), and negatively affect human health (Guo et al., 2017). Stagnant weather conditions and large anthropogenic emissions in this region are the main reasons for heavy haze and pollution (Zhao et al., 2013; Liu et al., 2013; Wang et al., 2013; Zhang et al., 2014; Zhang et al., 2015; Liu et al., 2017). Previous studies have shown feedbacks between aerosol and meteorological variables (e.g., Steiner et al., 2013; Tie et al., 2017; Huang et al., 2018; Li et al., 2018a; Wu et al., 2019). A high concentration of $PM_{2.5}$ can weaken turbulence (Ren et al., 2019) and enhance stability in the planetary boundary layer (PBL) resulting in decreased PBL height and consequently increased $PM_{2.5}$ concentrations (Su et al., 2018), i.e. a positive feedback between aerosols and PBL height (Petäjä et al., 2016). Liu et al. (2018) found another positive feedback: decreased PBL height can increase relative humidity (RH), and, in turn enhance secondary aerosol (SA) formation and further enhance particulate concentration, weaken solar radiation, and further decrease PBL height.

During severe haze events in the BTH region, $PM_{2.5}$ concentrations can increase as much as 200 $\mu g.m^{-3}$ in several hours (Zhong et al., 2017). On the one hand, SA formation through aerosol hygroscopic growth is one of main reasons causing this explosive growth of particular matter (Han et al., 2015; Sun et al., 2014; Chen et al., 2019). Huang et al. (2014), for example, found that severe haze pollution events were mostly the result of SA. On the other hand, vertical and horizontal transport can also produce an explosive growth of particular matter. Observations have shown that downward transport by large coherent eddies produced explosive growth of surface $PM_{2.5}$ concentrations (Han et al., 2018; Li et al., 2018b). Complex topography, land-use and land-cover, e.g. the Taihang and Yan Mountains, the Jing-Jin-Ji city cluster and Bohai Bay in BTH region, can also induce local circulations that affect air pollutant concentrations (Liu et al., 2009; Wang et al., 2017). The distribution of air pollutants in Beijing and Tianjin is readily modified by mountain-valley breezes and sea-land breezes. Regional transport by local-scale and weather-scale wind systems contribute significantly to the formation of severe haze events in the BTH region (Zheng et al., 2015; Sun et al., 2016; Ma et al., 2017; Jiang et al., 2015b). Dang et al. (2019) reported that regional transport was the most important process for the formation of severe winter haze days in the BTH region with a relative contribution of 65.3 %. Transport of aerosols to the BTH region by multi-scale circulations was also reported by Miao et al. (2017). However, due to a lack of dense vertical and horizontal spatial coverage of meteorological and aerosol observations, aerosol transport mechanisms and their effect on the explosive growth of particular matter in the BTH region are uncertain on both urban and smaller scales.

During a severe haze pollution event in 2015 in Beijing, a haze front (HF) phenomenon was reported on an internet social network, and also was intensively observed e.g. by scanning lidars and instrument networks during the 3-year field campaign of the Study of Urban Rainfall and Fog/Haze (SURF; Liang et al., 2018). The haze front resulted in $PM_{2.5}$ concentration increases of more than 200 μg m$^{-3}$ in a half hour or less when it passed over. This visible haze front in Beijing





resembles the smog front noted by Ahrens (2003) but differs in that its formation mechanism is
foehn winds. The foehn is characterized by a decrease in cloudiness on the lee side of mountains
and warm, dry, strong and gusty winds (Brinkmann 1971; Richner and Hächler 2013). The foehn
warming mechanisms are summarized into four types (Elvidge and Renfrew 2016): isentropic
drawdown, latent heating and precipitation, mechanical mixing due to turbulence, and radiative
heating. Foehns occur downstream of most major mountain ridges in the world (Drechsel and
Mayr 2008; Nkemdirim and Leggat 1978; Norte 2015; Takane and Kusaka 2011; Zhao et al.,
1993). Beijing city is located on the plains adjacent to the southwest-northeast oriented Taihang
Mountains and the northwest-southeast oriented Yan Mountains to the west and north, respectively.
Due to this topography, Beijing is often affected by foehns especially in the plains areas adjacent
to the mountains (Li et al., 2016, Wang et al., 2012a, 2012b). The temperature can increase sharply
in a short time: for example, Luo et al. has reported an intense foehn warming event in which
surface air temperature increased by more than 10℃ per hour in Beijing (Luo et al., 2020).
Despite the fact that the foehn has been investigated for a long time, there are few studies of the
influence of foehns on air pollution, especially in BTH region which has, for decades, been a
worldwide hotspot for studying air pollution (Gohm et al. 2009; Li et al. 2015; McGowan et al.
1996, 2002). Also, very few studies have reported foehn-related fronts. Vergeiner (2004) reported
a "minifront" or a convergence line between the up-valley flow and the down-valley foehn flow
that was maintained near the surface layer in the Wipp Valley, Austria. Li et al. (2015) found that a
ground-based foehn colliding with a thermally-driven valley breeze formed a "minifront" in
Urumqi, China, indicating that the foehn can play a critical role in the formation of severe air
pollution events in this area. Nonetheless, foehn-related haze fronts seem to have never been
investigated in the BTH region. Foehns usually come from western, northwestern and northern
mountain areas in the Heibei-Beijing region, which are usually less polluted than the plains area in
this region. Hence, foehns tend to increase visibility and decrease $PM_{2.5}$ concentrations in plains
areas (Yang et al., 2018). In our case study, the foehn initially decreased $PM_{2.5}$ concentrations in
the northern plains area of Beijing, and then interacted with a polluted air mass leading to severe
pollution in the urban area. This latter dramatic consequence has not been reported in previous
studies. The goal of our study is to investigate the structure and formation of this foehn-related
haze front on small scales in order to improve our understanding of the role of the foehn in air
pollution events in the BTH region. Intensive measurements from SURF as well as routine
measurements across the BTH region and historic data sets are described in Sect. 2. In Sect. 3, we
examine the characteristics and evolution of the haze front. In Sect. 4, we compare the formation
of a sea breeze front and a haze front, discuss the main reason for HF propagation and the role of
the foehn on air pollution. Sect. 5 is a summary.
2 Instrumentation, observations and data
Figure 1 and Table 1 shows the main observation sites mentioned here. The IAP site, which has
been operated by the Institute of Atmospheric Physics (IAP) since 1978, has a 325 m high
meteorological tower collocated with a ground-based Doppler lidar (Windcube 100S, Leosphere)
and a mini-micropulse aerosol lidar (mini-MPL, SigmaSpace). The Doppler lidar profiled the
mean wind using the Doppler beam swing (DBS) scan mode. The mini-MPL lidar profiled
aerosols and normalized relative backscatter (NRB). Wind direction and speed, air temperature,
and relative humidity were measured at 15 levels (8, 15, 32, 47, 65, 80, 100, 120, 140, 160, 180,



200, 240, 280, 320 m) on the tower. Three-dimensional sonic anemometers, downward- and
upward-pointing pyrgeometers and pyranometers (CNR1, Kipp & Zonen) and $CO_2/H_2O$
concentration sensors (LI-7500) were installed at 47 m, 140 m and 280 m. The instrumentation at
the IAP site is described in detail by Li et al. (2018b).
A mini-MPL lidar was installed at the top of the office building of the Institute of Urban
Meteorology (IUM), executing a plan-position indicator (PPI) scan mode at a 5 $^o$ elevation angle
and every 10 $^o$ of azimuth angle from 230 $^o$ to 340 $^o$ (Fig. 5d). An air quality station on the ground
~ 40 m south of the IUM office building provided 5-minute mean $PM_{2.5}$ concentrations. A weather
camera facing west was also installed in a room on the 9$^{th}$ floor of the office building taking
photos every 30 s. Due to the absence of a surface weather station at the IUM site, meteorological
data every 5 minutes from an automated weather station (AWS) at the Chedougou (CDG) site, ~ 1
km north of the IUM site, was jointly analyzed with data at IUM. An operational wind profiler
was installed at the Haidian (HD) National Weather station site. We also used radio soundings
launched twice daily at the Guanxiangtai (GXT) site. Hourly observations of $PM_{2.5}$ from all the
air quality stations were obtained from the website of the Ministry of Ecology and Environmental
of the People's Republic of China (http://106.37.208.233:20035). We used hourly $PM_{2.5}$ data at
Aotizhongxin (AOT) site, ~3 km northeast of the IAP site, and Yizhuang (YZ) site, ~4 km
northeast of the GXT site, to roughly represent $PM_{2.5}$ concentrations at IAP and GXT, respectively.
At the Changping (CP) site, the AWS of the Beijing Meteorology Service (BMS) is very close to
the air quality station of Beijing Municipal Ecological Environment Bureau (BMEEB). During the
study period, the Himawari satellite provided cloud images over the Beijing area every 10 minutes.
The pictures of this infrequent haze front were initially released by internet social communities,
e.g. SINA Weibo (equivalent to Chinese Twitter). Based on time and location information, we
selected two photos shown in Figure 1. All the observation heights used in this study are from
ground level (AGL). Also, in order to investigate foehn occurrence frequency and its relationship
to $PM_{2.5}$ concentrations in Beijing, 1 year AWS data at CP, AOT and GXT, and $PM_{2.5}$ data at CP,
AOT and YZ from 1 March 2015 to 29 February 2016 are used.
3 The evolution and characteristics of the haze front
3.1 Regional background of air pollution and weather conditions
This HF occurred on 24 December 2015 concurrent with a severe air pollution episode. The mean
$PM_{2.5}$ concentrations in Beijing varied between 300-400 μg m$^{-3}$ in the morning on 23 December, a
severe Air Quality Index (AQI) pollution level (Fig. 2a). Thereafter, the $PM_{2.5}$ concentration
decreased to ~100 μg m$^{-3}$ at 08:00 LST on 24 December and increased steadily up to 500 μg m$^{-3}$
by afternoon of the next day. A 500 mb trough passed Beijing at 08:00 LST on 24 December (Fig.
3a). Winds were predominantly westerly or northwesterly at ~ 500 m AGL (Figure 2a, 2b, 2d). On
the surface, a weak cold front was west of Beijing. On the right-side of the cold front, fog was
reported in Hebei and Shandong Provinces, and easterly flow was reported in Beijing by surface
meteorological stations. At 20:00 LST, there was a weak surface high centered north of Beijing.
The pressure gradient was weak with weak southwesterly surface flow in Beijing (Fig. 2c, Fig.
3f).
3.2 The evolution of the HF
The visible channel images from the Himawari satellite clearly showed the movement and
evolution of the HF as well as fog—dense white fog covered northeastern Tianjin and half of



Xianghe county of Heibei Province at 08:00 LST (Fig. 4a). Meanwhile, the Beijing area was clear
with low pollution. Southwest-northeast oriented clouds partly shadowed the fog in Tianjin and
Bohai Bay. Left of the fog front, the gray-white shading indicated hazy air with its front extending
just into the boundary of Beijing. The edge of the hazy air mass corresponding to the HF line
began to impact the GXT site (the blue dot, Fig. 4b, Fig. 11) at 10:00 LST followed by expanding
fog areas. The HF line moved slowly to the northwest while fog areas shrank quickly due to
increasing solar radiation. A west-east oriented high cloud street overlapped the hazy and foggy
areas (Fig 4d). After 16:00 LST, the fog disappeared and the HF line subsequently impacted the
FS, CBD and IUM sites (Fig. 1, Fig. 5), leaving a smaller unpolluted urban area on the
northwestern plains area in Beijing.
The Min-MPL at IUM scanned the HF passage using the PPI mode (Fig. 5c). The lidar initially
detected a hazy air mass to the southwest far away from the lidar site. As the HF approached, the
outline of the polluted air mass was clearly visible against the sky and buildings on the weather
camera photos (t1, Fig. 5e). When the HF arrived at the IUM site, the building view was blurred
by the hazy air mass (t2, Fig. 5e). Surface wind direction changed suddenly from NNW to WSW.
The $PM_{2.5}$ concentration jumped from ~10 to 269 μg m$^{-3}$ in 10 minutes (Fig. 5b). The wind
direction suddenly changed to northerly at 16:30 LST resulting in an abrupt $PM_{2.5}$ concentration
decrease to 11 μg m$^{-3}$ (Fig. 5b) and the visibility increased as evidenced by some visible buildings
(t3, Fig. 5e). In less than 10 minutes, the wind direction reverted again from N to NNW resulting
in a $PM_{2.5}$ concentration increase to 106 μg m$^{-3}$ (Fig. 5b) and again blurred the building view at
16:39 LST (t4, Fig. 5e). The scans showed five pollution 'waves' successively touched down at
the site and consequently led to severe pollution at 19:30 LST at IUM (Fig. 5b, 5c). The IAP site,
8 km northeast of IUM, was affected by the hazy air mass at around 20:30 LST according to
vertically scanning lidar observations at this site (Fig. 5a).
Based on the dense AWSs and air quality monitoring station coverage, we were able to analyze
surface distribution patterns of air temperature and humidity as they were affected by air flows and
$PM_{2.5}$ concentrations in the plains areas. The HF line was identified by temperature and humidity
contrasts between the warm and cold air masses and the convergence line of the surface wind field
(Fig. 6-7, Fig. S1-S3), which was also consistent with the front edge of the hazy air mass seen in
the satellite images (Fig. 4). The warm, dry, and clean air mass with more northerly winds was
surrounded by the cold, wet, and polluted air mass with more southerly or southeasterly winds
(Fig. 6-7). The position of the HF line oscillated due to the collision between the two air masses.
The HF line slowly advanced northwesterly with the southern part moving at about 2.5 km h$^{-1}$ (Fig.
6). The $PM_{2.5}$ contrast between the two air masses was more than 200 μg m$^{-3}$. At 16:00 LST, the
west-east oriented HF line crossed the main urban area and reached the IUM site. Later that night,
the wet and hazy air mass overlay most of the plains except for a small area on the northwestern
plains adjacent to the mountains (e.g., the CP site). As the foehn began to decrease and retreat, and
radiative heating decreased in late afternoon, the warm-dry air mass became weaker and shrank,
leading to dissolution of the HF. After the northerly gusty winds decreased after sunset, the
polluted air mass moved quickly toward the relative warm-dry and clean areas, resulting in severe
pollution over most of the plains areas in Beijing.

3.3 Characteristics of the HF and foehn winds



We used the CP, CDG and GXT sites (locations in Fig. 1) to investigate characteristics before and
after the hazy air mass passed through. The northernmost site CP was mostly unaffected by the
hazy air mass during 24 December. The southernmost site GXT was affected by the hazy air mass
the earliest at 10:00 LST on 24 December. Afterward, the $PM_{2.5}$ concentration at GXT varied from
349 to 515 μg m$^{-3}$ until midnight (Fig. 8e). The $PM_{2.5}$ concentration at CP was the lowest among
three sites with a maximum of 148 μg m$^{-3}$ at 11:00 LST and a minimum of 26 μg m$^{-3}$ at 15:00 LST.
The three-hourly temperature tendency showed that air temperature at CP increased significantly
at 11:00 LST due to the foehn; in contrast, air temperatures decreased significantly at CDG and
GXT. Meanwhile, humidity decreased and wind speed increased at CP due to the foehn. The warm
and dry foehn wind was initially detected over the northwestern mountains and plains of
Changping County at around 11:00 LST, with a significant increase in temperature and the north
wind component, and decrease in humidity. Wind profiler observations at HD also showed
enhanced upper-air winds. From 10:00 to 13:00 LST, air temperature increased from 1.9℃ to 6℃,
relative humidity decreased from 49% to 24%, and wind speed increased from 1 m/s to 4.6 m/s at
CP. The foehn affected CDG at about 12:00 LST and IAP at 13:00 LST before colliding with a
cold, wet, and hazy air mass. At 11:00 LST, an urban heat island (UHI) formed in the main urban
areas mainly due to intense solar heating under a clear and clean sky but also due to the heat
released by urban activities. At 12:00 LST, the warm-dry air mass driven by gusty the foehn
merged with the UHI, enlarging the coverage of the warm air mass. When the HF passed over
CDG, the pressure significantly increased (~0.5 hpa, Fig. 8a) as well as humidity and $PM_{2.5}$
concentrations, but temperature slightly decreased (Fig. 8b). CDG was also affected by the foehn
at around 12:00 LST, 1.5 hours later than CP, with accompanying temperature and wind speed
increases, and decrease in humidity.
At IAP around noon, a northwesterly wind and an updraft increased significantly between 450
m and 1250 m height above the surface, and the wind direction below 500 m changed from
northeast to northwest (Fig. 9). Concurrently, the tower temperatures also significantly increased
and relative humidity decreased, and the wind profiles changed below 320 m (Fig. 10a-b), which
implies IAP was affecting by the foehn at this time. The temperature increased mainly from 12:00
LST to 19:00 LST when turbulence also increased significantly (Fig. 10c). As the HF approached,
the wind weakened, the wind direction changed to southwest, and the humidity increased sharply.
The atmosphere became more stably stratified near the surface, leading to enhanced pollution
below the hundred meter level. Figure 11 shows downward shortwave radiation (DR) at IAP and
GXT, and $PM_{2.5}$ concentrations at AOT and YZ. Aerosols reduced the DR to 225 W m$^{-2}$ at GXT at
11:00 LST, 36 W m$^{-2}$ less than IAP. The radiation difference between IAP and GXT was 174 W
m$^{-2}$ at 13:00 LST. Meanwhile, the $PM_{2.5}$ concentration was 456 μg m$^{-3}$ greater at YZ than ATZX.
At GXT, higher concentrations of aerosol particles in the polluted air mass scattered more solar
radiation and reduced the amount of solar radiation at the ground, leading to weaker turbulence
and lower PBL height which further enhanced the aerosol concentration near the ground. In
contrast at IAP, there were less aerosol particulates, more radiation and stronger turbulence
resulting in higher PBL height and less air pollutant concentration near the ground.
4 Discussion
The formation of the HF is illustrated in Figure 12. The fundamental process of HF formation is
similar to the concept of colliding density currents illustrated by Simpson (1997) and Kingsmill et





al. (2003), i.e, the collision of a gust front with a sea-breeze front (SBF). This results in high
concentrations of pollutants in the convergence zone of the front (Yoshikado et al., 1996; Dong et
al., 2017; Li et al., 2019). As noted by Miller et al., (2003), who described the structure and
characteristics of the SBF in detail, the sea breeze occurs under relatively cloud-free skies, when
the surface of the land heats up more rapidly than the sea. But in our case, the coastal area of
Bohai Bay was covered mostly by clouds, fog, and haze during the daytime (Fig. 4), which
decreased the contrast between the land and the sea and inhibited the sea breeze. Also, sea breezes
occur normally around 11:00 LST in the Bohai Bay area in winter, which is later than in summer
(Qiu and Fan, 2013). The typical speed of a SBF is 3.4 m s$^{-1}$ (Simpson et al., 1977), considerably
larger than the observed HF speed of ~0.7 m s$^{-1}$. Again, the front we studied was not a SBF;
nevertheless, the SBF has some similarities in shape and formation to the HF in our study. The sea
breeze is one example of a gravity or density current, which are primarily horizontal flows
generated by a density difference of only a few percent. Field studies have confirmed that the SBF,
as part of the sea breeze gravity current, has aspects of the sea breeze head (SBH), which had been
simulated by Simpson (1994, 1997) using laboratory tanks and two bodies of water of slightly
different densities. Likewise, in our case, a temperature difference between the warm air mass and
the cold air mass resulted in a density difference between these air masses. Figures S4-S7 show air
density distribution overlapping surface wind vectors and PM$_{2.5}$ concentrations. Note in these
figures that the biggest gradient of air density corresponds to the wind convergence line as well as
the HF line. Both the radiative heating difference between northern and southern areas, and the
warm and dry foehn are key factors producing warm air masses. Southern hazy air masses reflect
more solar radiation, and thus inhibit an increase in surface temperature and turbulence mixing,
leading to colder and denser air. In contrast, solar radiative heating enhances heating of the
northerly clear and cleaner air mass which concurrently was affected by the warm foehn, leading
to warmer and less dense air. The lower density air mass collides with the higher density air mass
and subsequently overrides the denser air mass (Fig. 1c; Fig. 5e; Fig. 12). This aspect is very
similar to the SBF. The warmer air overriding the cold-wet air mass also intensifies the inversion
at the top of the boundary between two air masses, limiting the growth rate of the underlying layer
and increasing its pollutant concentration. The interface between the warm-dry air mass and the
cold-wet air mass formed the HF and a significant convergence line at the surface. This kind of
convergence line can sometimes be found during air pollution events when pollutants transferred
by southerly winds encounter northerly mountain winds at night in the BTH region (Li et al, 2019;
Liang et al., 2018).
Typically for clear daytime conditions, the horizontal temperature differences between the air
over mountains and the adjacent plains can produce up-slope, up-valley, and plains-mountain
winds, which are usually weak and often overpowered by strong foehns (Whiteman, 2000) and
intensified wind speed in the upper air (Fig. 2d) as is the case here. Our results also show that the
pressure difference between air masses is about 0.5 hpa before and after the HF passage (Fig. 8).
This pressure gradient forcing creates a seesaw clash between the polluted and the clean air
masses. The polluted air mass repeatedly encroaches into the clean air mass and is pushed back by
the clean air mass. Eventually the polluted air mass wins. The lidar observed five wave-like
polluted air invasions (Fig. 5). Thus the pressure difference between the air masses due to different
air densities, although small, caused the hazy air mass to slowly swing north or northwest, and
inflicted severe pollution on the urban area. Li et al., 2016 showed that the polluted aerosol



concentration in southern Beijing is normally higher than in the urban and northern rural areas of
Beijing (Li et al., 2016). For typical regional air pollution events in Beijing, air pollutants are
mainly transported from surrounding areas, especially Hebei Province and Tianjin, south of
Beijing (Zheng et al., 2015; Zheng et al., 2018). Both urban heat island effects and aerosol
radiation forcing result in polluted areas that are colder with higher air density than less polluted
areas, leading to a weak pressure gradient between more polluted and less polluted air masses.
During the daytime, the pressure gradient forcing by air density is overlapped by the forcing of
plains-mountain winds, valley breezes and urban heat island circulations, enhancing air pollutant
transport from southern to northern areas. At night, if mountain-plains winds and mountain
breezes are weak, the pressure gradient forcing by air density can transport polluted air toward less
polluted areas, which seems not to have been discussed previously. It implies that the weak
pressure gradient could play an important role on air pollutants transport during the weak
mountain-plain wind system and mountain-valley breeze periods.

In order to investigate the occurrence frequency of the foehn in Beijing, we analyzed one year
of AWS data (PM2.5 data) at three sites CP (CP), AOT (AOT) and GXT (YZ), representing
northern suburban, urban and southern suburban areas, respectively (Table 2). For daily data
sampled hourly, if meteorological variables of the CP site at one hour meet the following criteria:
(1) one-hour temperature increase is the highest among the three sites and greater than 1.5℃ and
at least 1.0℃ higher than that at the GXT site, (2) one-hour relative humidity tendency is negative,
and (3) hourly wind direction is greater than 270 ° or less than 90 °, we define this day as a foehn
case day. These critera ensure that the CP site adjacent to the mountains has the most significant
warming among three sites with the foehn coming from the mountain and with a relative humidity
decrease, i.e. a typical foehn case. Note in Table 2 that during the months from OCT to MAR
when severe haze events are also frequent in Beijing, there is a higher foehn frequency than other
months. There are 16 foehn cases, about 55% of all cases, connected to air pollution events. In 10
cases, PM2.5 concentrations for all sites decreased 24 hrs after the foehn's occurrence. In 1 case,
PM2.5 concentrations for all sites increased after the foehn's occurrence. In 5 cases, including the
case in this article, PM2.5 concentrations for all sites initially decreased then increased 24 hrs after
the foehn's occurrence. The foehn's effects on air pollution can be direct or indirect. The direct
effect is that gusty foehns transport air pollutants resulting in increasing or decreasing air pollution
concentration depending on whether the foehns are polluted or clean. The more complicated
indirect effect alters air mass properties and boundary layer structure by dry and warm foehn
winds, which then influences air pollution. It is worth noting that type B cases in Table 2,
accounting for 17% of total foehn cases, are likely due to the indirect effect leading to heavier air
pollution, and need to be investigated further.

5 Summary and implications
This is the first study to our knowledge in which an HF related to the foehn in the BTH region has
been analyzed. Based on observations collected during SURF-15, we studied an HF on 24
December 2015 in Beijing. This HF was formed by the collision between a cold-humid polluted
air mass with higher density and a warm-dry clean air mass with lower density which was mainly
due to the foehn. Initially, fog occupied the plains areas southeast of Beijing associated with a
hazy air mass early in the morning. The hazy-foggy air mass developed and invaded Beijing
around noon. The fog dissipated in the afternoon. The warm-dry downslope foehn began to impact



the northern plains before noon and moved to the south, gradually affecting other plains sites until colliding with the cold-wet polluted air mass south and east of the urban areas, leading to a convergence line and the HF boundary at noon. As the HF passed by surface sites, $PM_{2.5}$ concentrations increased by more than 200 μg m$^{-3}$ in 10 minutes. Following the HF, four surges of polluted air invaded the IUM site and consequently produced severe pollution. The HF boundary was clearly visible from satellite images and weather camera photos during the daytime. The formation of the HF is very similar to the SBF, although the front in our study cannot be explained as an SBF due to weak radiation and temperature contrast between the land and sea, its earlier occurrence time and long distance inland. The sloped boundary of the HF tilts toward the polluted air mass as a result of the warm-dry clean air mass overriding the cold-wet polluted air mass. The HF slowly swings toward northern and northwestern clean areas. Our results show that as the foehn wind weakened and retreated, the weak pressure gradient between the warm-dry air mass and the cold-wet air mass was the main factor forcing the polluted air mass to slowly move north or northwest.

We segregate the effect of the foehn on air pollution into two types: a direct and an indirect effect. This foehn-induced HF event gives us a good opportunity to investigate both direct and indirect effects of the foehn on air pollution and haze events. Some studies have revealed the direct effect of the foehn on air pollution: stronger gusty foehns could diminish or even eliminate air pollutants. For the seldom-studied indirect effect of the foehn, it could enhance differences in radiation and air density between clean and polluted air masses, resulting in a weak pressure gradient between air masses which allows the polluted air mass to invade the clean air mass. This mechanism could be more dominant especially when upslope winds, valley winds, and plains-mountain winds retreat after sunset in Beijing. Also, warm-dry foehns could affect urban heat island and atmosphere stratification in the boundary layer, and further affect air pollution.

Although air pollution events in BTH region have been studied from different aspects over decades, few studies have investigated the influence of the foehn on air pollution. Therefore, we recommend further studies on the formation mechanism of the foehn and its effects on air pollution in the BTH region.

*Data availability*. The $PM_{2.5}$ data is available on the website of Ministry of Ecology and Environmental of the People's Republic of China (http://106.37.208.233:20035). Other data are available at http://www.ium.cn/dataCenter/ which archives the SURF filed data collected from 2015 and 2016.

*Author contributions*. JL, MZ had the original idea, JL, ZS, YD, ZC, YW and QL performed the integrative data analysis, JL, ZS, and DL wrote the manuscript. All authors discussed the results and commented on the paper.

*Competing interests*. The authors declare that they have no conflict of interest.

*Acknowledgments*. The authors would like thank the anonymous reviewers for their helpful comments. This work was supported by the National Natural Science Foundation of China (41875123); the Ministry of Science and Technology of China (Grant No. 2016YFC0203302); Beijing Natural Science Foundation of China (8171002). This material is based upon work





supported by the National Center for Atmospheric Research, which is a major facility sponsored
by the National Science Foundation under Cooperative Agreement No. 1852977.





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



**Table 1.** Latitude, longitude, altitude, site type and instrumentation at the main sites in Fig.1.

| Site name | Latitude | Longitude | Altitude (m MSL) | Site type | Instrumentation |
|---|---|---|---|---|---|
| GXT | 39.806 | 116.469 | 31 | Meteorology | radiosonde, radiometer, wind profiler, AWS, pyranometer |
| IAP | 39.974 | 116.371 | 29 | Meteorology | 325-m tower, sonic anemometers, doppler lidar, vertical scanning MPL, pyrgeometer and pyranometer |
| IUM | 39.95 | 116.28 | 37 | Meteorology | PPI scanning MPL, weather camera, particulates sampler and analyzer |
| CDG | 39.945 | 116.291 | 58 | Meteorology | AWS |
| CP | 40.223 | 116.212 | 76 | Meteorology & Air quality | AWS, particulates sampler and analyzer |
| HD | 39.987 | 116.291 | 46 | Meteorology | AWS, wind profiler |
| AOT | 39.982 | 116.397 | – | Meteorology & Air quality | particulates sampler and analyzer |
| YZ | 39.795 | 116.506 | – | Air quality | particulates sampler and analyzer |


※ AWS: automated weather station; MPL: micro pulse lidar; PPI: plain position indicator






**Table 2.** Number of foehn cases in Beijing from 1 March 2015 to 29 February 2016

|  |  | JAN | FEB | MAR | APR | MAY | JUN | JUL | AUG | SEP | OCT | NOV | DEC | Total cases |
|---|---|---|---|---|---|---|---|---|---|---|---|---|---|---|
| Clean cases |  | 4 | 2 | 1 | 0 | 1 | 1 | 0 | 1 | 0 | 2 | 1 | 0 | 13 |
| Polluted cases | Type A | 1 | 2 | 2 | 0 | 1 | 0 | 0 | 0 | 0 | 2 | 0 | 2 | 10 |
|  | Type B | 1 | 0 | 1 | 0 | 0 | 0 | 0 | 0 | 0 | 0 | 1 | 2 | 5 |
|  | Type C | 0 | 0 | 0 | 0 | 0 | 0 | 0 | 0 | 1 | 0 | 0 | 0 | 1 |
| All cases |  | 6 | 4 | 4 | 0 | 2 | 1 | 0 | 1 | 1 | 4 | 2 | 4 | 29 |

Clean cases: foehn cases in which average $PM_{2.5}$ concentrations for CP, AOT and YZ is less than 50 μg m$^{-3}$.
Polluted cases: foehn cases in average $PM_{2.5}$ concentrations for CP, AOT and YZ is greater than 50 μg m$^{-3}$.
Type A: polluted cases with $PM_{2.5}$ concentrations for CP, AOT and YZ sites decreasing since foehn's occurrence in 24 hours.
Type B: polluted cases with $PM_{2.5}$ concentrations for CP, AOT and YZ sites initially decreasing then increasing since foehn's
occurrence in 24 hours.
Type C: polluted cases but $PM_{2.5}$ concentrations for CP, AOT and YZ sites increasing since foehn's occurrence in 24 hours.
All cases: all of clean cases and polluted cases.







Figure Captions


Figure 1. (a) Locations of main observational stations (red dots: meteorological stations; blue dots:
air quality monitoring stations) used here, along with the fourth, fifth and sixth ring
roads (blue lines) in Beijing, China. Anonymous photos from SINA Weibo (like Twitter
but in Chinese) at (b) Financial Street (FS) at 15:30 and (c) Central Business District
(CBD) at 16:00 on 24 December 2015. Locations for taking the photos are shown as
yellow stars in the upper map.
Figure 2. (a) Hourly-mean PM2.5 concentration of 35 air quality stations in Beijing from 23
December to 25 December. Temperature (solid), dew point (dashed), and wind vectors
(in knots) from the radio sounding profiles at (b) 00:00 UTC and (c) 12:00 UTC at GXT.
(d) The temporal variation of hourly wind-vector profiles from the wind profiler at HD
on 24 December 2015.
Figure 3. The 500 mb weather maps at (a) 08:00 LST and (b) 20:00 LST, the 800 mb weather
maps at (c) 08:00 LST and (d) 20:00 LST, and surface maps at (e) 08:00 LST and (f)
20:00 LST on 24 December 2015.
Figure 4. Cloud images from the visible channels of the Himawari satellite at (a) 08:00 LST, (b)
10:00 LST, (c) 12:00 LST, (d) 14:00 LST, (e) 15:00 LST, and (f) 16:00 LST on 24
December 2015. Red, yellow and blue dots are the locations of IUM, IAP and GXT,
respectively.
Figure 5. (a) The Min-MPL vertically scanned normalized relative backscatter (NRB) at IAP
station. (b) 5-min average wind speed (red line), wind direction (triangles) at CDG
station and PM2.5 concentrations (blue line) at IUM station. (c) The NRB from a
Min-MPL at IAP scanning in (d) plan position indicator (PPI) mode using 10o
horizontal angle intervals from 340o to 220o and with the elevation angle at 5o. (e) Four
photos taken by the weather camera at IUM at times t1, t2, t3 and t4 which are marked
as red lines in the upper plot.
Figure 6. Temperature (color filled dots coded according to the bottom color bar), wind vectors (2
m s-1 is one full bar) and PM2.5 concentrations (purple circles; the size of the circle
represents concentration values) in the plain area within and around Beijing at (a) 12:00
LST and (b) 16:00 LST. The blue lines indicate the location of the haze front.
Figure 7. Same as Fig 6, but with relative humidity.
Figure 8. Three-hourly (a) pressure tendency, (b) air temperature tendency, and (c) specific
humidity, (d) wind speed (lines) and wind directions (colored dots), and (e) PM2.5
concentration at CP (red line), CDG (blue dash line) and GXT (black dash line) on 24
December 2015.
Figure 9. Doppler lidar observations of (a) vertical wind velocity, (b) horizontal wind speed, (c)
wind direction, and (d) carrier-noise-ratio (CNR) at IAP on 24 December 2015.
Figure 10. Temporal variations of (a) temperature (colored contours) and wind vectors, (b) relative
humidity (colored contours) and wind vectors at 15 levels on the IAP tower, and (c)
vertical velocity variance at 47 m and 280 m on the IAP tower on 24 December 2015.
Figure 11. The temporal variations of PM2.5 concentrations at ATZX (blue bars) and YZ (red
bars), and downward short-wave radiation at GXT and at heights of 47 m, 140 m and
280 m at the IAP tower during daytime on 24 December 2015.
Figure 12. Schematic diagram of the haze front formation.

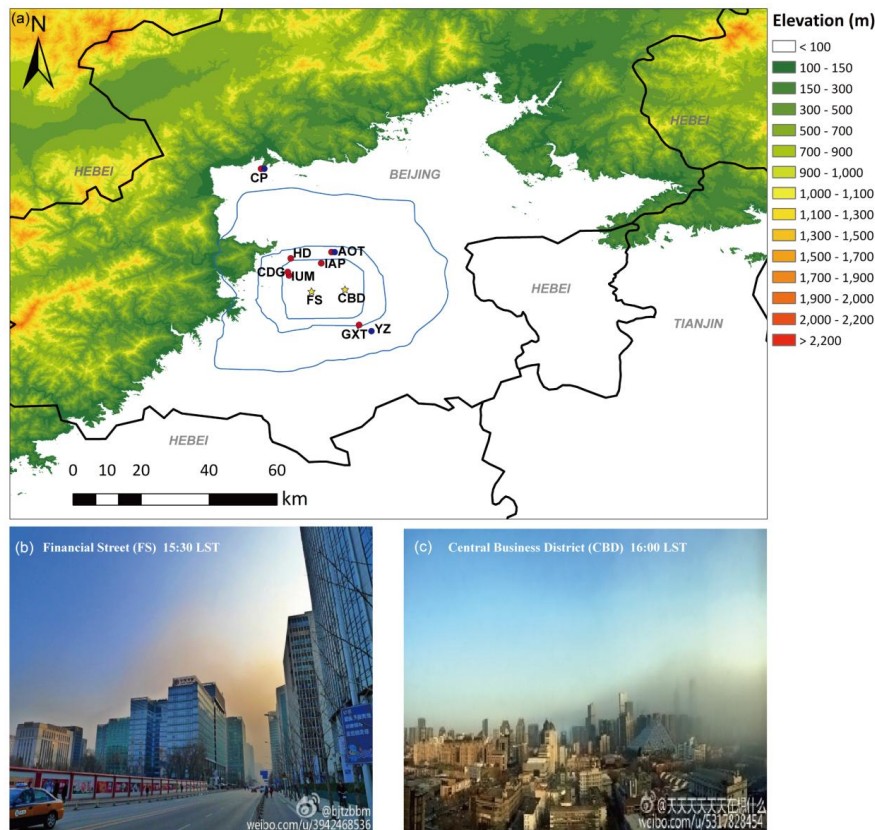

Figure 1. (a) Locations of main observational stations (red dots: meteorological stations; blue dots:
air quality monitoring stations) used here, along with the fourth, fifth and sixth ring roads (blue
lines) in Beijing, China. Anonymous photos from SINA Weibo (like Twitter but in Chinese) at (b)
Financial Street (FS) at 15:30 and (c) Central Business District (CBD) at 16:00 on 24 December
2015. Locations for taking the photos are shown as yellow stars in the upper map.



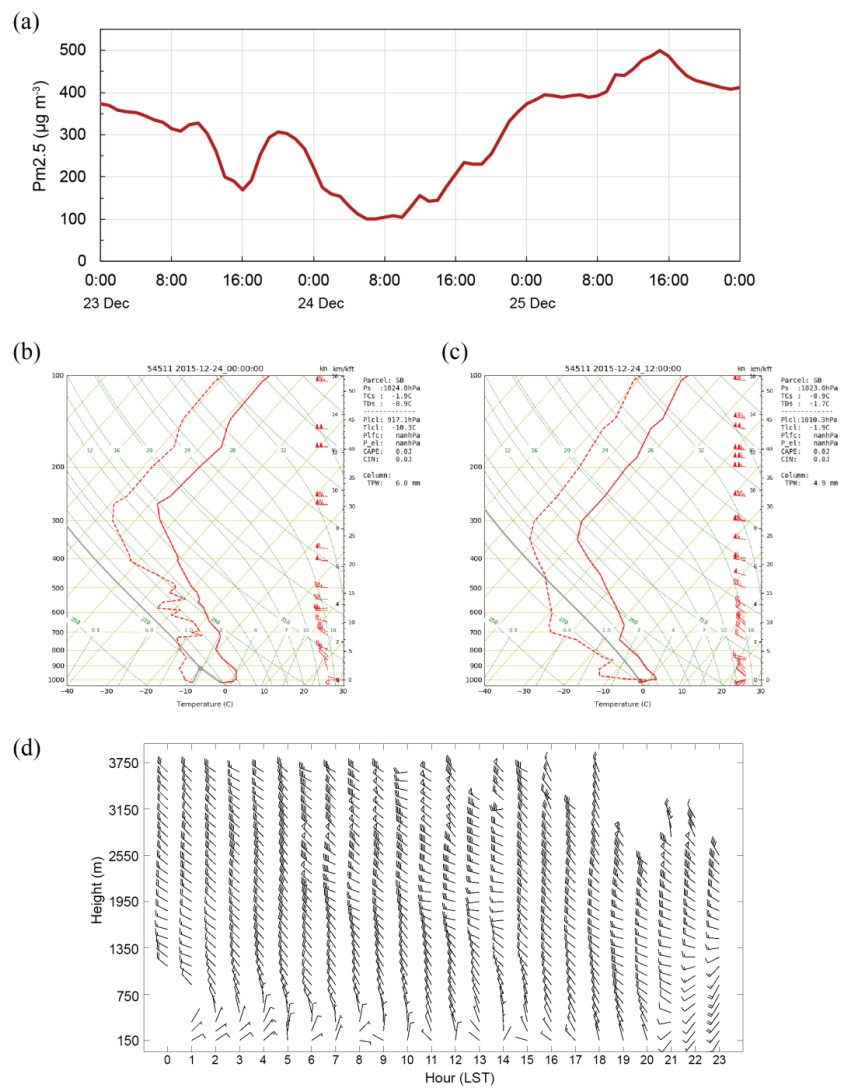


Figure 2. (a) Hourly-mean PM$_{2.5}$ concentration of 35 air quality stations in Beijing from 23

December to 25 December. Temperature (solid), dew point (dashed), and wind vectors (in knots)

from the radio sounding profiles at (b) 00:00 UTC and (c) 12:00 UTC at GXT. (d) The temporal

variation of hourly wind-vector profiles from the wind profiler at HD on 24 December 2015.




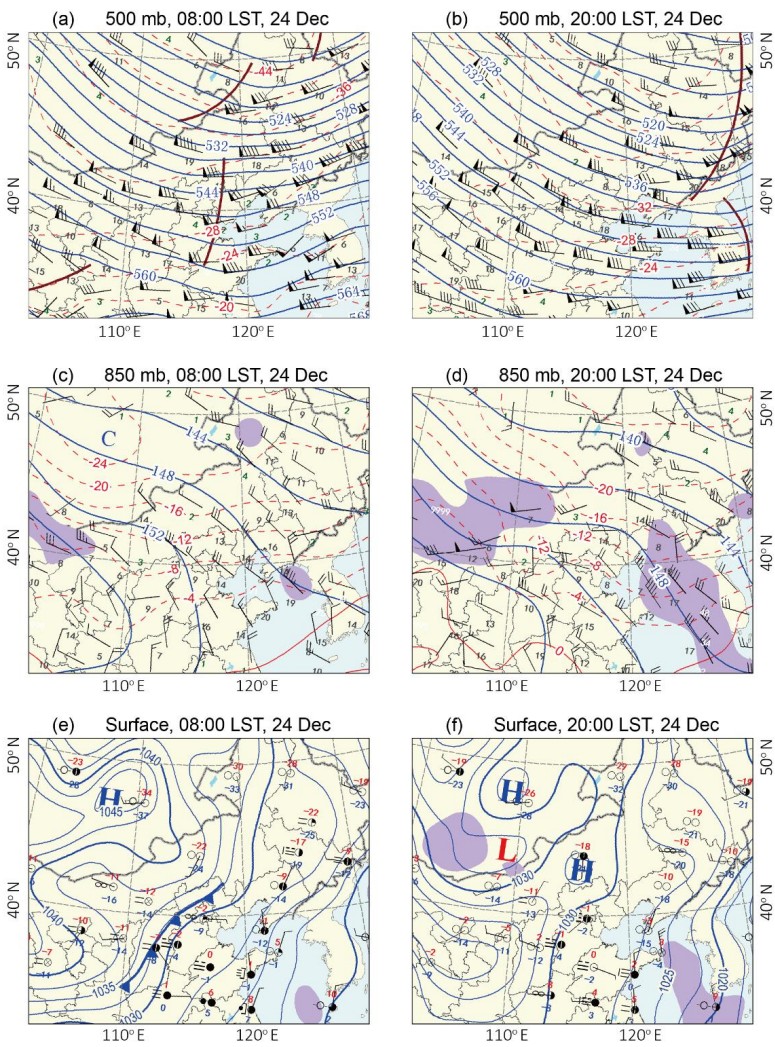


Figure 3. The 500 mb weather maps at (a) 08:00 LST and (b) 20:00 LST, the 800 mb weather
maps at (c) 08:00 LST and (d) 20:00 LST, and surface maps at (e) 08:00 LST and (f) 20:00 LST
on 24 December 2015.





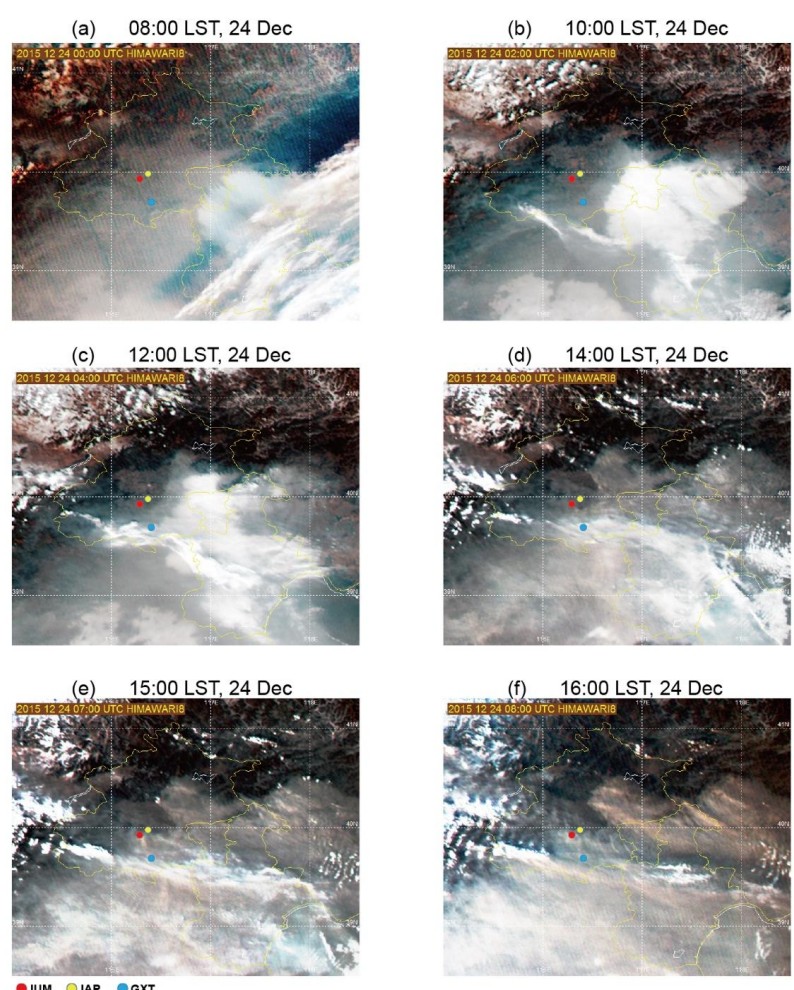

Figure 4. Cloud images from the visible channels of the Himawari satellite at (a) 08:00 LST, (b)
10:00 LST, (c) 12:00 LST, (d) 14:00 LST, (e) 15:00 LST, and (f) 16:00 LST on 24 December 2015.
Red, yellow and blue dots are the locations of IUM, IAP and GXT, respectively.

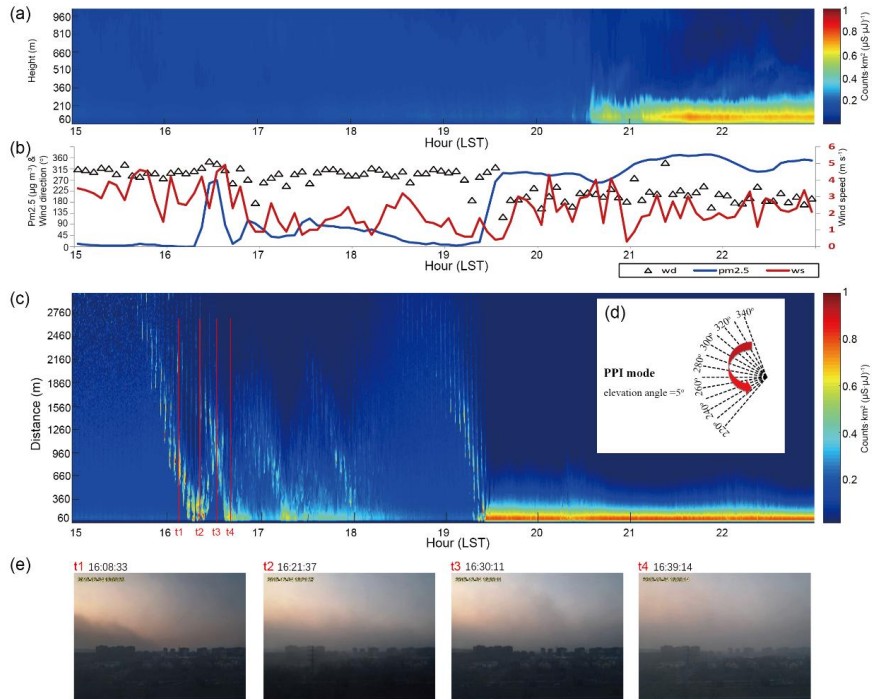


Figure 5. (a) The Min-MPL vertically scanned normalized relative backscatter (NRB) at IAP
station. (b) 5-min average wind speed (red line), wind direction (triangles) at CDG station and
PM$_{2.5}$ concentrations (blue line) at IUM station. (c) The NRB from a Min-MPL at IAP scanning in
(d) plan position indicator (PPI) mode using 10° horizontal angle intervals from 340° to 220° and
with the elevation angle at 5°. (e) Four photos taken by the weather camera at IUM at times t1, t2,
t3 and t4 which are marked as red lines in the upper plot.



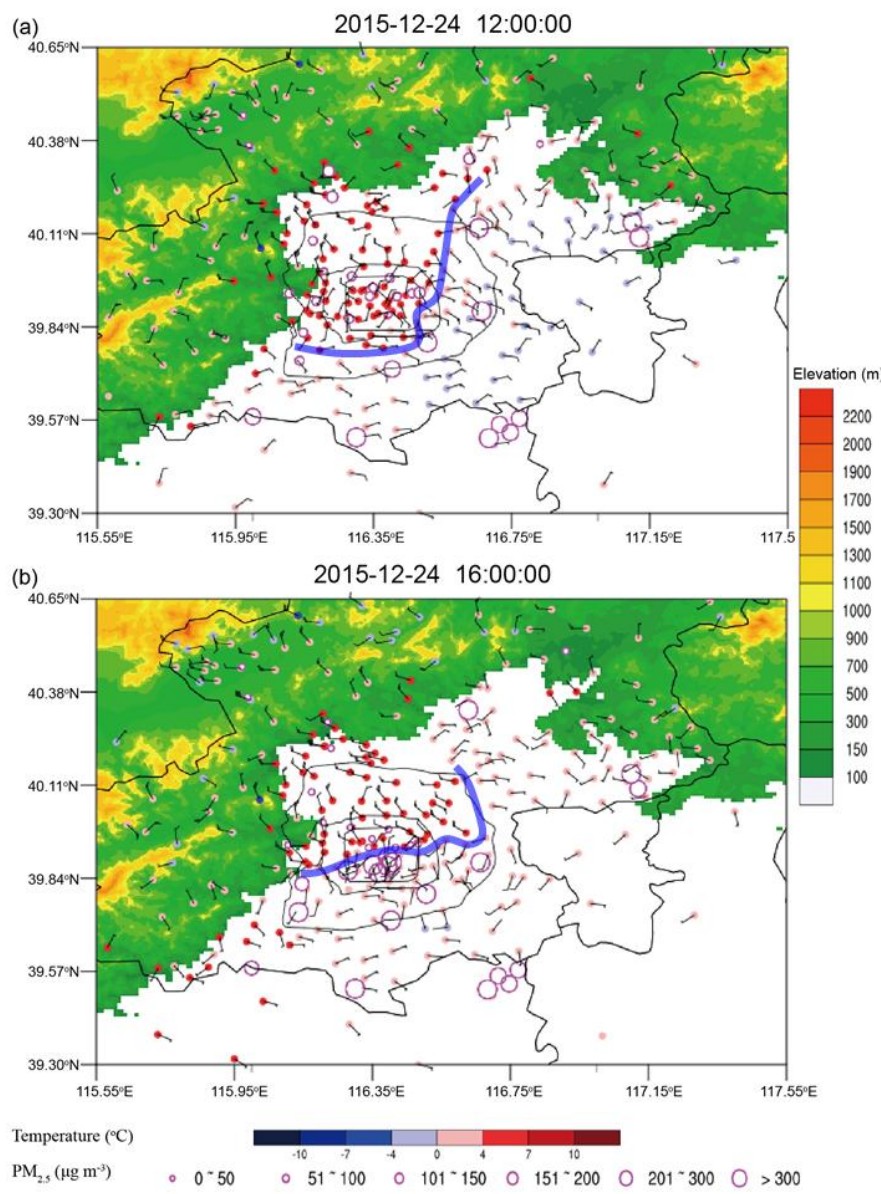

Figure 6. Temperature (color filled dots coded according to the bottom color bar), wind vectors (2
m s$^{-1}$ is one full bar) and PM$_{2.5}$ concentrations (purple circles; the size of the circle represents
concentration values) in the plain area within and around Beijing at (a) 12:00 LST and (b) 16:00
LST. The blue lines indicate the location of the haze front.



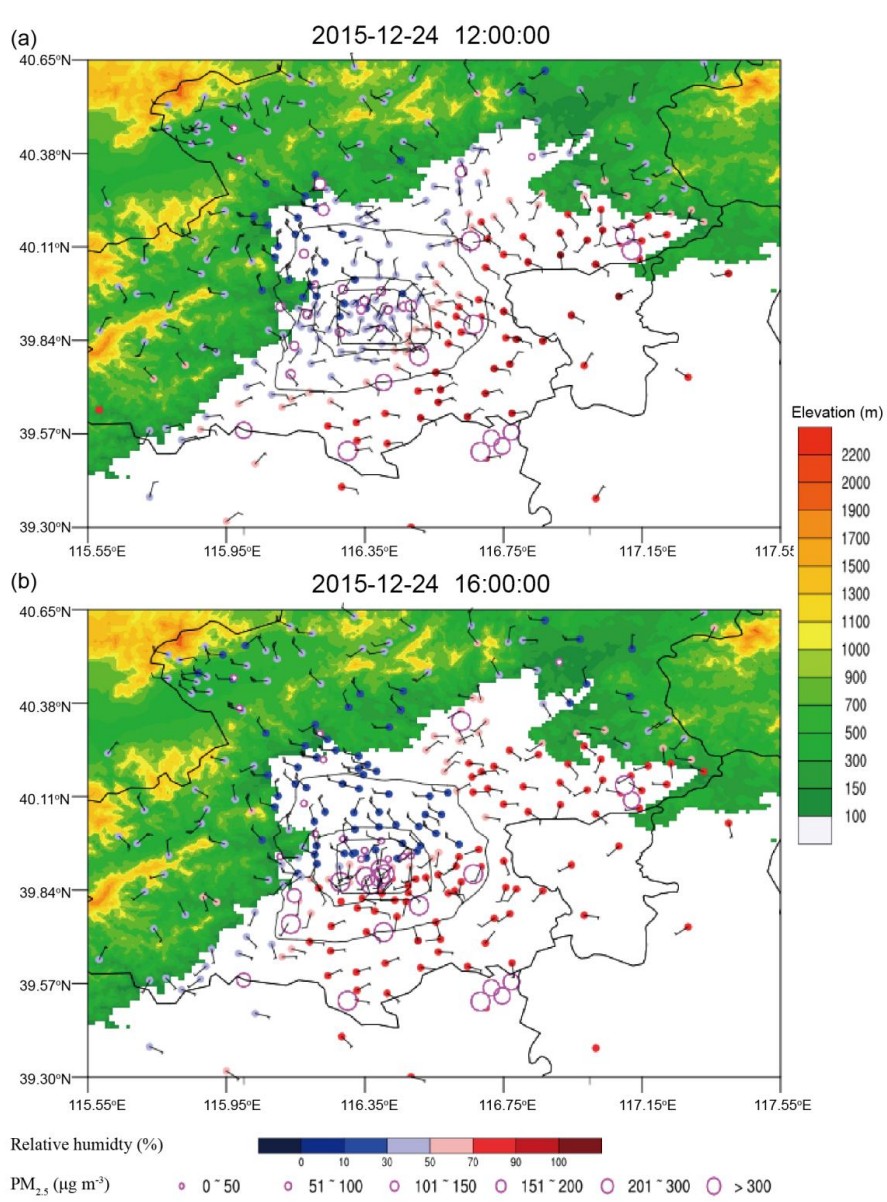

Figure 7. Same as Fig 6, but with relative humidity.

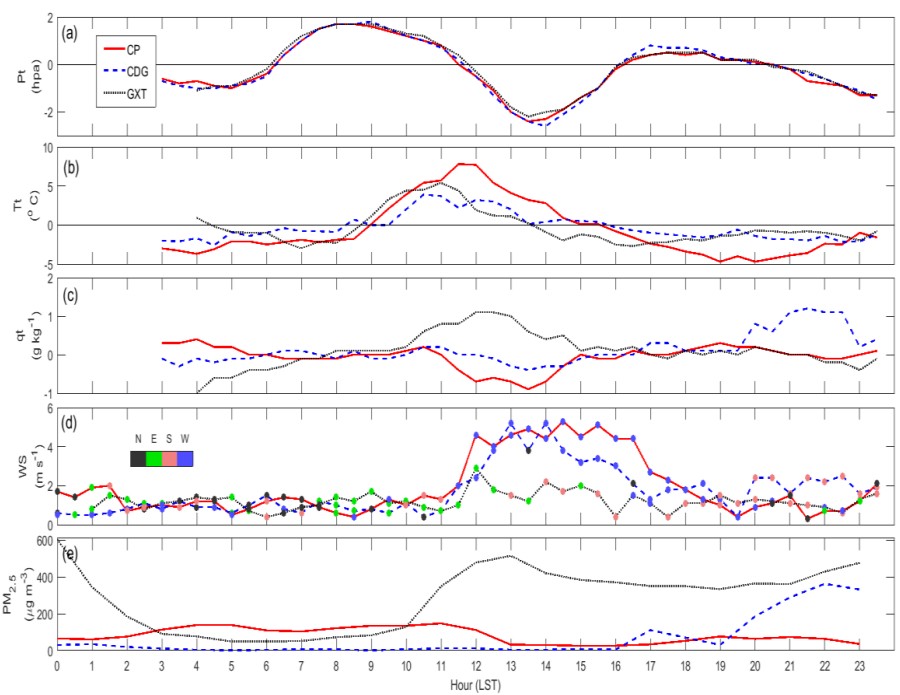


Figure 8. Three-hourly (a) pressure tendency, (b) air temperature tendency, and (c) specific
humidity, (d) wind speed (lines) and wind directions (colored dots), and (e) PM$_{2.5}$ concentration at
CP (red line), CDG (blue dash line) and GXT (black dash line) on 24 December 2015.






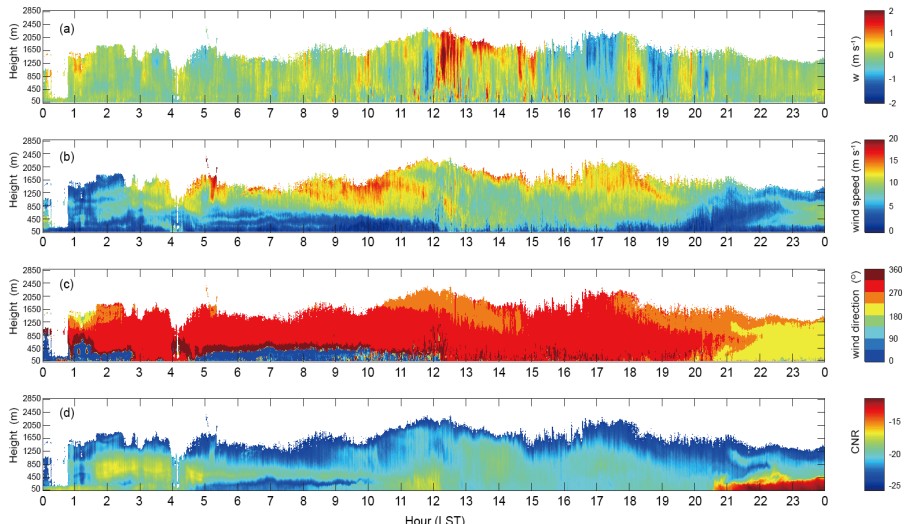


Figure 9. Doppler lidar observations of (a) vertical wind velocity, (b) horizontal wind speed, (c)
wind direction, and (d) carrier-noise-ratio (CNR) at IAP on 24 December 2015.


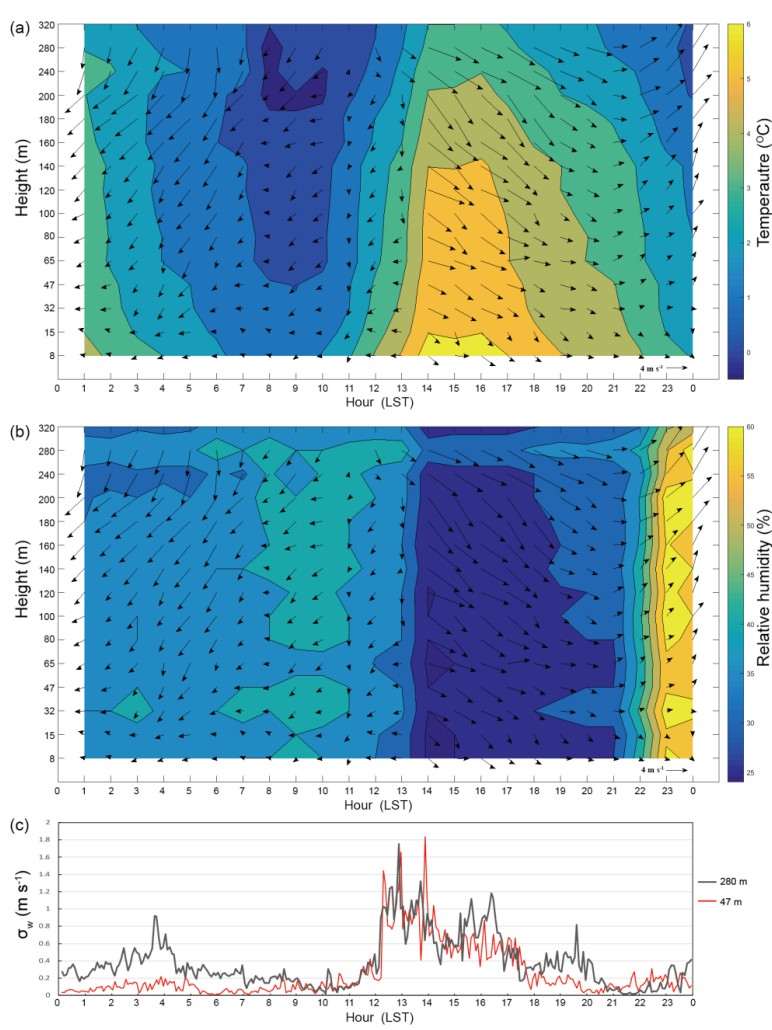


Figure 10. Temporal variations of (a) temperature (colored contours) and wind vectors, (b) relative
humidity (colored contours) and wind vectors at 15 levels on the IAP tower, and (c) vertical
velocity variance at 47 m and 280 m on the IAP tower on 24 December 2015.





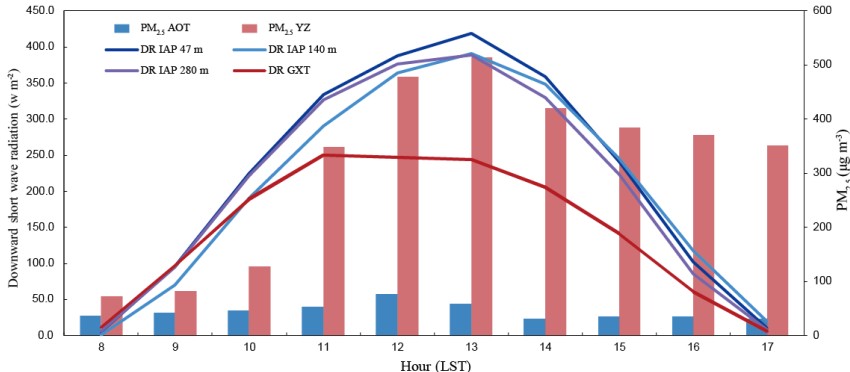


Figure 11. The temporal variations of PM$_{2.5}$ concentrations at ATZX (blue bars) and YZ (red bars),
and downward short-wave radiation at GXT and at heights of 47 m, 140 m and 280 m at the IAP
tower during daytime on 24 December 2015.





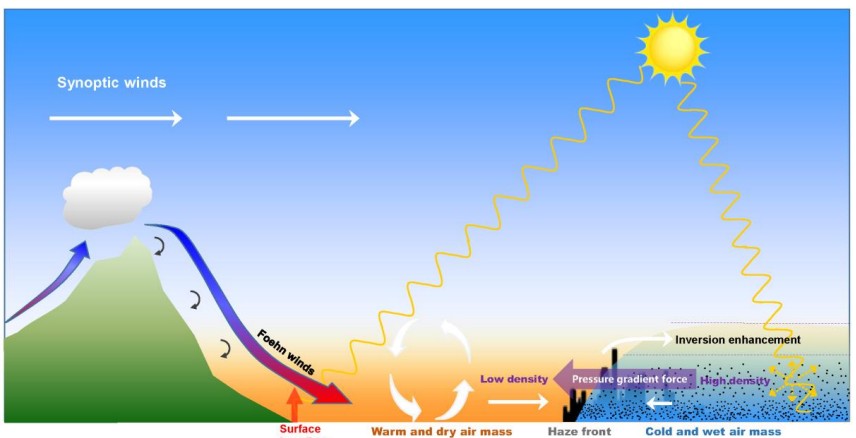

Figure 12. Schematic diagram of the haze front formation.