# Peer review of "A foehn-induced haze front in Beijing: observations and implications"

_Atmospheric Chemistry and Physics, 2020_

## Referee Comment (RC1) · Anonymous Referee #1 · 25 Sep 2020

General comments:

The manuscript discusses a pollution event that produced a "haze front" over Beijing, China on 24 December 2015. The haze front is a sharp change in visibility on the boundary of warm dry clean air meeting cool moist polluted air over the city. The warm air mass was created by a foehn wind blowing down from the mountains to the north and west of Beijing. The interaction between the two air masses is analysed in detail using meteorological variables, and particular matter measurements from a number of sites across the city. The authors then conclude with statistics of how often this sort of event occurs. These events can have a large impact on air quality in Beijing and can potentially be applied to other cities with a similar geographical layout. I am not aware of a similar publication into haze fronts caused by foehn winds.

[Figure]

The manuscript reads well and the English is good, but there are some sections that need changing. Great use of many different measurement platforms. As there are a lot of sites, Table 1 and the map in Figure 1 really help for orientation. But some of the figures need some work (see below). The supplemental figures are helpful for providing a broader picture. I could not find a video (mentioned during the submission).

Specific comments:

Define the time zone LST (I assume it's local time)

$PM_{2.5}$: sometimes the 2.5 is subscript or normal script

Mini-MPL: sometimes you refer to the Mini-MPL as Min-MPL

Table 1:

- observation heights missing

- AOT AWS is missing

Section 3.1 needs to discuss the meteorological conditions in a bit more detail.

- You show a number of weather charts, but don't really explain what the relevance is to the haze front case study. The upper level trough is not really relevant here.

- You don't reference the 850 mb charts in the main text. What height is the 850 mb surface? This is useful when comparing to the wind profiler data (Fig. 2d). But ultimately the surface wind is weak and doesn't really correlate with the upper level data in the morning. Figure 10 also shows very low wind speeds in the morning before the foehn arrives. Looking at Figure S1a there doesn't seem to be any evidence of a cold front or dominant wind direction.

- Figure 3: if possible add some dots to show measurement sites if they are not too close together. Maybe same as Figure 4 (IUM, IAP and GXT). As a minimum a dot showing the centre of Beijing. Also, in the caption it should be '850 mb'.

- Line 158 says the flow is weak south-westerly in Figure 2c, but looks weak north-easterly to me in Figure 3f. Which one is it?

- The radiosonde profile in Figure 2c also shows up the vertical extent of the haze front at GXT nicely.

- Figure 2a: Is it possible to add some sort of distribution of hourly PM2.5 values from the different sites (percentiles)? The event is highly variable. Also, what area does the average PM2.5 cover?

Section 3.2:

- Figure 4: it might help to add a narrow line to the images where you can pick out the haze front as it is quite difficult to see at first (similar to Figure 6)

- Figure 5b: The text says that the wind direction changes suddenly at t2 (ie. 16:21 LST) (line 177) but I can't see this in the figure. Are the wind directions in the plot correct?

- Figure 5 (caption, line 673): The scanning lidar is at IUM not IAP.

- Figure 7: can you add the haze front line to this plot? Also, personal preference: maybe flip the colour bar (red as dry and blue as moist)

- line 193: revise the word "surrounded"

Section 3.3:

- Figure 8 has values every hour and not three-hourly as mentioned in the caption. The use of tendency is a bit confusing. Using it for pressure works, to show the small differences between sites that produce the pressure gradient. Are the tendencies subtracted from the mean of the whole day? Unless the absolute values of temperature and humidity for the different sites are very different, I would find it easier if they were absolute (or at least include the mean values in the plot eg. Tmean[CP]=x.xdegC) as you reference absolute values in the main text.

[Figure]

- line 223: I think "gusty the foehn" should be "the gusty foehn". Also, what do you mean by "enlarging the coverage"?

- line 225: do you mean "the pressure increased significantly compared to the other sites"?

- line 229: Reference that it is IAP Doppler lidar data in the text to make it clear. Otherwise I have to look at the figure to work that out.

- line 233: Please reword. I think you mean something like: "the temperature was higher and the turbulence was increased mainly between 12:00 LST and 19:00 LST". The current wording implies that it started low at 12:00 LST and increased steadily to a maximum at 19:00 LST.

- line 236 and Figure 10: you might benefit from including potential temperature here, as this shows up stability better.

- line 237: Looks like the enhanced pollution wasn't just below 100 metres. Figure 9 suggests the aerosol went all the way up to 400 metres.

- Figure 11: Are the tower plots correct? I'd expect DR to decrease as you get closer to the ground, but 47 m shows the highest DR. Also, it might be easier to read if you list the tower levels in one column in the legend.

- line 240 and Figure 11: do you mean AOT instead of ATZX?

- line 243: The lower PBL height is also to do with the PBL height being suppressed by the overrunning foehn wind, not just the reduced solar radiation at the ground.

Section 4:

- line 302: are mountain-plains winds and mountain breezes the same thing?

- line 319: add the types in brackets (eg. '(Type A)') to the main text to make it more clear.

- Table 2: rewording for the types: eg. "Type A: polluted cases where PM2.5 concentrations for the CP, AOT and YZ sites had decreased 24 hours after the foehn's occurrence."

Data availability:

Couldn't access http://www.ium.cn/dataCenter/ (Page not found)

Couldn't access http://106.37.208.233:20035/ (Couldn't install Silverlight on my system)

---

## Referee Comment (RC2) · Anonymous Referee #2 · 23 Oct 2020

This observational study describes the evolution of meteorological conditions and PM concentrations associated with a foehn event. As the warm, dry, and relatively clean foehn wind meets the cold, wet and polluted air mass in the Beijing area, a haze front formed. The large observational network captured the characteristics and evolution of this haze front as it moved through the network. The level of details captured by both the ground and the upper air observations (including wind profilers, Doppler Lidars and radiosounds) makes this study a useful contribution to the literature regarding foehn characteristics and its influence on air pollution.

The manuscript contains a large number of figures that are generally in good quality. The writing, however, could use substantial improvements, as described below.

Define the key terms. The central focus of the manuscript is on haze front, but there is

[Figure]

no clear definition on what a haze front is and how to identify it from the observations. It is stated in the paper that "The HF line was identified by temperature and humidity contrasts between the warm and cold air masses and the convergence line of the surface wind field". How does this differ from foehn front? Is HF the same as foehn front? I would expect that a HZ should be identified by sharp contrast in PM values, instead of temperature and humidity. If HZ and foehn front are the same, then say it. In sum, it should be clearly stated near the beginning what you mean by haze front, foehn front, and what criteria are used to identify these fronts from your observational data.

The manuscript could benefit from reorganization. It is good to begin with describing background conditions for this episode. The sequence of the current description is: PM time series, sounding profiles, profiler winds, and finally synoptic patterns. I would reverse the order, starting from synoptic patterns and ending with surface observations including PM time series. More details are needed in the description of synoptic conditions. Right now, the synoptic patterns are shown, but there is very little discussion. In addition to describing the synoptic patterns for this case, there should be some discussion on how typical the pattern is and how often it occurs in order to put this particularly episode into historical context. Further, I do not see the need to separate "The evolution of HF "and "Characteristics of the HF and foehn winds" into two sections.

Some of the discussion could be improved Some of the discussion is rather confusion. For example, "This HF occurred on 24 December 2015 concurrent with a severe air pollution episode." But according to the PM time series in Fig. 2a, PM is high on the 23rd, but dropped down to nearly 100 on the morning of the 24th, and gradually increased to nearly 500 in the afternoon of 25. So if HF occurred on the 24th, then it was not concurrent with severe air pollution episode. In fact, in the next paragraph, it is mentioned that "the Beijing area was clear with low pollution." It is unclear what the background pollution level was and what was associated by HF. Is the increase on the 24th due to the passage of HF? I would expect a sharp increase instead of a gradual increase. In the discussion of satellite images, it is unclear how you distinguish

haze from fog or clouds in the satellite images? Some of the descriptions used present tense while others used past tense. Be consistent. The discussion about the HF and foehn characteristics is exhaustion to read. Better rewriting is necessary to improve readability.

Figures Figures are generally in good quality, but figure captions could use more details. For example, the caption for Figure 1 should include a description of the different symbols, although they are described in the text. Also include AWS and PM stations in Figure 1.

The font size for the axis labels in the time series plots and some of text in the figures should be enlarged. They are currently too small to read unless the figures are enlarged by 200% (e.g., Fig. 10 c vertical axis, Fig. 10 b, the label for the color bar; Figure 2b).

Clearly mark the time of HF passage on the time series plots.

---

## Author Comment (AC1) · 5 Nov 2020

**Response to reviewer's comments**

Anonymous Referee #1:

General comments:

The manuscript discusses a pollution event that produced a "haze front" over Beijing, China on 24 December 2015. The haze front is a sharp change in visibility on the boundary of warm dry clean air meeting cool moist polluted air over the city. The warm air mass was created by a foehn wind blowing down from the mountains to the north and west of Beijing. The interaction between the two air masses is analysed in detail using meteorological variables, and particular matter measurements from a number of sites across the city. The authors then conclude with statistics of how often this sort of event occurs. These events can have a large impact on air quality in Beijing and can potentially be applied to other cities with a similar geographical layout. I am not aware of a similar publication into haze fronts caused by foehn winds. The manuscript reads well and the English is good, but there are some sections that need changing. Great use of many different measurement platforms. As there are a lot of sites, Table 1 and the map in Figure 1 really help for orientation. But some of the figures need some work (see below). The supplemental figures are helpful for providing a broader picture.

We sincerely thank the reviewer for in-depth comments and helpful suggestions. We have responded to all the comments point-by-point and made corresponding changes in the manuscript. Following are detailed responses to all the comments.

I could not find a video (mentioned during the submission).

We are very sorry for forgetting to upload the video during the submission. This video will be submitted as a supplement along with the revised manuscript.

Specific comments:

Define the time zone LST (I assume it's local time)

The time zone LST (Local Standard Time) is Beijing time. We have defined the time zone LST in the manuscript.

$PM_{2.5}$: sometimes the 2.5 is subscript or normal script

Thank you. All of 2.5 in $PM_{2.5}$ have been set as subscripts in the manuscript.

Mini-MPL: sometimes you refer to the Mini-MPL as Min-MPL

Thank you. All 'Min-MPL' has been changed to 'Mini-MPL'.

Table 1:

- observation heights missing

Thank you. This has been corrected.

- AOT AWS is missing

Thank you. This has been corrected.

Section 3.1 needs to discuss the meteorological conditions in a bit more detail.
- You show a number of weather charts, but don't really explain what the relevance is to the haze front case study. The upper level trough is not really relevant here.

Thank you. We have added more explanations to this section. We also have adjusted the sequence of Section 3.1 as Referee #2 suggested. In brief, the upper air flows were dominated by northwesterly winds. And before the impact of the synoptic system, surface winds were weak. This synoptic pattern was one of typically frequent unfavorable conditions which exacerbate air pollution.

- You don't reference the 850 mb charts in the main text. What height is the 850 mb surface? This is useful when comparing to the wind profiler data (Fig. 2d). But ultimately the surface wind is weak and doesn't really correlate with the upper level data in the morning. Figure 10 also shows very low wind speeds in the morning before the foehn arrives. Looking at Figure S1a there doesn't seem to be any evidence of a cold front or dominant wind direction.

The height of 850 mb at Beijing is ~ 1500 m above the ground level. The winds at this height in Figure 2d agree well with the winds at same height in Figure 3c and 3d. We agree with you that the cold front marked in Figure 3e, which was automatically plotted by the software, seems not obvious and unwarranted. Therefore, we have re-plotted the map without the cold front.

- Figure 3: if possible add some dots to show measurement sites if they are not too close together. Maybe same as Figure 4 (IUM, IAP and GXT). As a minimum a dot showing the centre of Beijing.

Thank you for the comment. It is hard to distinguish sites in such small maps. So we have made the boundaries of Beijing bold and brown on the maps.

Also, in the caption it should be '850 mb'.

The caption has been corrected.

- Line 158 says the flow is weak south-westerly in Figure 2c, but looks weak northeasterly to me in Figure 3f. Which one is it?

The flow is southwesterly in Figure 3f. Please see an enlarged figure below. The black dot pointed by a red arrow is the GXT site with ~ 1 m/s southwesterly surface wind.

[Figure]

- The radiosonde profile in Figure 2c also shows up the vertical extent of the haze front at GXT nicely.

Thank you for the comment.

- Figure 2a: Is it possible to add some sort of distribution of hourly PM2.5 values from the different sites (percentiles)? The event is highly variable.

We are not sure about how to plot the percentiles in Figure 2. An alternative way is to plot hourly PM2.5 value for all three representative sites and three-sites mean value. Please see the new Figure 2a.

Also, what area does the average PM2.5 cover?

The average PM2.5 is the mean PM2.5 concentration of 35 sites in Beijing shown on the below site map and the new Figure 1a. Most of the sites are located in plain areas in Beijing.

[Figure]

Source: http://zx.bjmemc.com.cn/getAqiList.shtml?timestamp=1601452931296

Section 3.2:

- Figure 4: it might help to add a narrow line to the images where you can pick out the haze front as it is quite difficult to see at first (similar to Figure 6)

Thank you for the comment. We have added red lines to illustrate the haze front in Figure 4.

- Figure 5b: The text says that the wind direction changes suddenly at t2 (ie. 16:21 LST) (line 177) but I can't see this in the figure. Are the wind directions in the plot correct?

The wind directions in the plot are correct. As a result of using wind data observed at Chedougou (CDG) site, ~ 1 km north of the IUM, the wind direction changes at CDG didn't match the PM2.5 concentration changes at IUM very well. In fact, there is an AWS collocated with the Mini-MPL at the top of the IUM building, which observed wind direction and wind speed but with more missing data during the HF passing period. We used wind direction and wind speed data at IUM and redrew the Figure 5b. Besides the wind data observed at the IUM site, you can also find the wind direction changes agreed well with the PM2.5 concentration changes from the supplemental video.

- Figure 5 (caption, line 673): The scanning lidar is at IUM not IAP.
Thank you. This has been corrected.

- Figure 7: can you add the haze front line to this plot? Also, personal preference: maybe flip the colour bar (red as dry and blue as moist)
Thank you for the comment. We have added the haze front line to Figure 7, and flipped the color bar as you suggested.

- line 193: revise the word "surrounded"
Thank you for the comment. We have revised the word "surrounded". The sentence now reads:
*"The foehn winds with the warm, dry, and clean air collided with more southerly or southeasterly winds with the cold, wet, and polluted air and resulted in oscillations of the HF line (Fig. 6-7)."*

Section 3.3:
- Figure 8 has values every hour and not three-hourly as mentioned in the caption. The use of tendency is a bit confusing. Using it for pressure works, to show the small differences between sites that produce the pressure gradient. Are the tendencies subtracted from the mean of the whole day? Unless the absolute values of temperature and humidity for the different sites are very different, I would find it easier if they were absolute (or at least include the mean values in the plot eg. Tmean[CP]=x.xdegC) as you reference absolute values in the main text.

Thank you for the comment. The pressure tendency we used here is defined as $P_{t0}$-$P_{t0-3}$, where $P_{t0}$ is the pressure at t0 (current time in hour) and $P_{t0-3}$ is the pressure at 3 hours before t0. A similar tendency definition was used for the temperature and specific humidity. We agree with your comment that the use of tendency is a bit confusing. Hence, we have re-plotted Figure 8a, 8b and 8c by using the half-hourly pressure anomaly, temperature and specific humidity, respectively. We have marked the time of HF passage at CDG in Figure 8 as Refree #2 suggested. Also, we have revised sentences related to the changes in Figure 8.

- line 223: I think "gusty the foehn" should be "the gusty foehn".

Thank you. This has been corrected.

Also, what do you mean by "enlarging the coverage"?

By "enlarging the coverage", we mean to express the enlarged scope or coverage of the UHI after the warm-dry air mass driven by the gusty foehn merged with the UHI.

- line 225: do you mean "the pressure increased significantly compared to the other sites"?

We meant the pressure increased significantly when the haze front passed over the CDG site. But because we modified Figure 8 and found the pressure only increased slightly, we have revised the sentence. It now reads:

*"When the HF passed over CDG, the humidity and PM2.5 concentrations significantly increased, the pressure slightly increased, but the temperature slightly decreased (Fig. 8)."*

- line 229: Reference that it is IAP Doppler lidar data in the text to make it clear. Otherwise I have to look at the figure to work that out.

Thank you for the comment. We have revised the sentence. The sentence now reads:

*"At IAP around noon, the Doppler lidar detected a northwesterly wind and a significantly increased updraft between 450 m and 1250 m height above the surface, and the wind direction below 500 m changed from northeast to northwest (Fig. 9)."*

- line 233: Please reword. I think you mean something like: "the temperature was higher and the turbulence was increased mainly between 12:00 LST and 19:00 LST". The current wording implies that it started low at 12:00 LST and increased steadily to a maximum at 19:00 LST.

Thank you. We have reworded the sentence.

- line 236 and Figure 10: you might benefit from including potential temperature here, as this shows up stability better.

Thank you for the comment. But there are no pressure observations at 15 levels on the IAP tower.

- line 237: Looks like the enhanced pollution wasn't just below 100 meters. Figure 9 suggests the aerosol went all the way up to 400 meters.

Thank you. We reworded the sentence. The sentence now reads:

*"The boundary layer became more stably stratified near the surface, leading to enhanced pollution in the lowest few hundred meters."*

- Figure 11: Are the tower plots correct? I'd expect DR to decrease as you get closer to the ground, but 47 m shows the highest DR.

Thank you for the comment. The tower plots are correct.

The reason that the DR at 47 m is highest is due to the 'canyon trapping' effect on shortwave radiation in the urban canopy layer (Oke, 2017). The rugosity of the urban surface contributes to the lower albedo values at higher solar elevation angles as more short-wave radiation enters

the street canyons where it is trapped (Christen and Vogt, 2004; Kotthaus and Grimmond, C.S.B., 2013). The IAP tower is surrounded by 4- to 20-story buildings with heights of 10 to 60 m (Liu et al., 2017; Wang et al., 2019). Some buildings just south of the IAP tower are higher than the pyrgeometers and pyranometers mounted at 47 m on the tower causing the additional shortwave radiation trapping in the street canyon, especially when solar elevation angles are greater (Dou et al., 2018).

Also, it might be easier to read if you list the tower levels in one column in the legend.

Thank you for the comment. We have listed the tower levels in one column in the legend.

- line 240 and Figure 11: do you mean AOT instead of ATZX?

Thank you. Yes. We have changed all "ATZX" into "AOT".

- line 243: The lower PBL height is also to do with the PBL height being suppressed by the overrunning foehn wind, not just the reduced solar radiation at the ground.

Thank you for the comment. According to our observational analysis, there is no evidence that foehn winds affected GXT. Please see in Figure S2a the southern most areas that foehn winds have invaded. Hence, the lower PBL height at GXT is mainly due to less solar radiation and less turbulence, and positive feedback between the PBL height and the aerosol concentration.

Section 4:

- line 302: are mountain-plains winds and mountain breezes the same thing?

Mountain breezes are a part of mountain and valley breezes that refer to the cooler, more-dense air that glides downslope into the valley during nighttime (Ahrens, 2003). Mountain-plain winds result from horizontal temperature differences between the air over a mountain massif and the air over the surrounding plains, producing large-scale winds that blow up or down the outer slopes of a mountain massif (Whiteman, 2000). The scale of mountain-plain winds is larger than the scale of mountain breezes.

- line 319: add the types in brackets (eg. '(Type A)') to the main text to make it more clear.

Thank you. We have reworded the sentences.

- Table 2: rewording for the types: eg. "Type A: polluted cases where PM2.5 concentrations for the CP, AOT and YZ sites had decreased 24 hours after the foehn's occurrence."

Thank you for the comment. We have reworded the table note.

Data availability: Couldn't access http://www.ium.cn/dataCenter/ (Page not found)

Sorry about the invalid link. The link of the datacenter of IUM has been changed into http://www.ium.cn:8088/ in 2020.

Couldn't access http://106.37.208.233:20035/ (Couldn't install Silverlight on my system)

Please visit https://quotsoft.net/air/ where the PM2.5 data sources from the China Environmental Monitoring Station and the Beijing Environmental Protection Testing Center.

References for review replies:

Ahrens C. D.: Meteorology Today: An Introduction to Weather, Climate and the Environment. Thomson Learning, USA, 2003.

Christen, A., Vogt, R.: Energy and radiation balance of a Central European City. Int. J. Climatol. 24, 1395–1421, 2004.

Dou J, Grimmond S, Cheng Z, Miao S, Feng D, Liao M: Summertime surface energy balance fluxes at two Beijing sites. IntJ Climatol., 39, 2793–2810, https://doi.org/10.1002/joc.5989, 2019.

Kotthaus, S., Grimmond, C.S.B. Energy exchange in a dense urban environment – Part II: Impact of spatial heterogeneity of the surface. Urban Climate, 10, 261-280, http://dx.doi.org/10.1016/j.uclim.2013.10.001, 2013.

Liu, J. K., Gao, Z. Q., Wang, L. L., Li, Y. B., and Gao, C. Y.: The impact of urbanization on wind speed and surface aerodynamic characteristics in Beijing during 1991–2011, Meteorol. Atmos. Phys., 130, 311–324, https://doi.org/10.1007/s00703-017-0519-8, 2017.

Oke, T.R., Mills, G., Christen, A. and Voogt, J.A.: Urban Climates. Cambridge: Cambridge University Press, 525 pp. https://doi.org/10.1017/9781139016476, 2017.

Wang, L., Liu, J., Gao, Z., Li, Y., Huang, M., Fan, S., Zhang, X., Yang, Y., Miao, S., Zou, H., Sun, Y., Chen, Y., and Yang, T.: Vertical observations of the atmospheric boundary layer structure over Beijing urban area during air pollution episodes, Atmos. Chem. Phys., 19, 6949–6967, https://doi.org/10.5194/acp-19-6949-2019, 2019.

Whiteman, C. D.: Mountain Meteorology: Fundamentals and Applications, Oxford Univ. Press, New York, 2000.

---

## Author Comment (AC2) · 5 Nov 2020

**Response to reviewer's comments**

Anonymous Referee #2:

This observational study describes the evolution of meteorological conditions and PM concentrations associated with a foehn event. As the warm, dry, and relatively clean foehn wind meets the cold, wet and polluted air mass in the Beijing area, a haze front formed. The large observational network captured the characteristics and evolution of this haze front as it moved through the network. The level of details captured by both the ground and the upper air observations (including wind profilers, Doppler Lidars and radiosounds) makes this study a useful contribution to the literature regarding foehn characteristics and its influence on air pollution.

The manuscript contains a large number of figures that are generally in good quality. The writing, however, could use substantial improvements, as described below.

We sincerely thank the reviewer for in-depth comments and helpful suggestions. We have responded to all the comments point-by-point and made corresponding changes in the manuscript. Following are detailed responses to all the comments.

Define the key terms. The central focus of the manuscript is on haze front, but there is no clear definition on what a haze front is and how to identify it from the observations. It is stated in the paper that "The HF line was identified by temperature and humidity contrasts between the warm and cold air masses and the convergence line of the surface wind field". How does this differ from foehn front? Is HF the same as foehn front? I would expect that a HZ should be identified by sharp contrast in PM values, instead of temperature and humidity. If HZ and foehn front are the same, then say it. In sum, it should be clearly stated near the beginning what you mean by haze front, foehn front, and what criteria are used to identify these fronts from your observational data.

Thank you for the comment. A haze front is not the same as a foehn front, but has some similarity to a foehn front in some cases, like in the case we studied here. "The haze front" is denoted mainly by its front-like structure and sharp contrast in polluted aerosol. This is why we didn't use the name "foehn front" in the manuscript. The foehn front is basically the foehn-induced "minifront" phenomena described in the literature by Vergeiner (2004) and Li et al. (2015). We identify a haze front by sharp contrast in PM values, instead of temperature and humidity. Thus, we have stated the meaning of the haze front in the introduction. The new added sentence reads:

*"This HF was identified by a sharp contrast in PM$_{2.5}$ concentration and a convergence line in the surface wind field."*

We also have revised the sentence you quoted above. The sentence now reads:

*"The HF line was identified by a sharp contrast in PM$_{2.5}$ concentration, temperature and humidity between the warm and cold air masses and the convergence line of the surface wind field (Fig. 6-7, Fig. S1-S3), which was also consistent with the front edge of the hazy air mass seen in the satellite images (Fig. 4)."*

The manuscript could benefit from reorganization. It is good to begin with describing background conditions for this episode. The sequence of the current description is: PM time series, sounding profiles, profiler winds, and finally synoptic patterns. I would reverse the order, starting from synoptic patterns and ending with surface observations including PM time series. More details are needed in the description of synoptic conditions. Right now, the synoptic patterns are shown, but there is very little discussion. In addition to describing the synoptic patterns for this case, there should be some discussion on how typical the pattern is and how often it occurs in order to put this particularly episode into historical context.

Thank you for the comment. We have adjusted the sequence of Section 3.1 as you suggested. Also, we have added more description of synoptic conditions in this section, and compared this synoptic pattern with the statistics of unfavorable synoptic conditions which exacerbate air pollution (Wang et al., 2020).

Further, I do not see the need to separate "The evolution of HF "and "Characteristics of the HF and foehn winds" into two sections.

Thank you for the comment. We have combined the section "3.2 The evolution of HF" and the section "3.3 Characteristics of the HF and foehn winds" into one section, named "3.2 The evolution and characteristics of the HF and foehn winds".

Some of the discussion could be improved. Some of the discussion is rather confusion. For example, "This HF occurred on 24 December 2015 concurrent with a severe air pollution episode." But according to the PM time series in Fig. 2a, PM is high on the 23rd, but dropped down to nearly 100 on the morning of the 24th, and gradually increased to nearly 500 in the afternoon of 25. So if HF occurred on the 24th, then it was not concurrent with severe air pollution episode. In fact, in the next paragraph, it is mentioned that "the Beijing area was clear with low pollution." It is unclear what the background pollution level was and what was associated by HF. Is the increase on the 24th due to the passage of HF? I would expect a sharp increase instead of a gradual increase.

Thank you for the comment. Because of using the mean $PM_{2.5}$ concentration of 35 sites in Beijing in Figure 2a, it is hard to distinguish the highly varied air pollution on 24th. So we have replaced Figure 2a with a new figure using hourly-mean $PM_{2.5}$ concentration of CP, AOT, YZ and 3-stations mean. According to the new Figure 2a, the Beijing area was clear with low pollution on the morning of the 24th, which was also verified from satellite images (Figure 4a, 4b). In brief, the background pollution level was low in Beijing before the passage of the HF on the 24th. The HF affected YZ causing a sharp increase in PM concentration at around noon. But meanwhile, $PM_{2.5}$ concentration decreased inversely at CP and kept a very low level at AOT. When the HZ arrived at AOT at 22:00 LST, it caused a sharp increase in PM (increased 268 μg m$^{-3}$ in one hour). Consequently, the PM increase on the 24th was mainly due to the passage of HF. We have rewritten this whole paragraph. The sentences now read:

*"This HF occurred on 24 December 2015 after a severe air pollution event. The mean $PM_{2.5}$ concentrations of CP, AOT and YZ varied between 300-400 μg m$^{-3}$ on the morning of 23 December, which is a severe Air Quality Index (AQI) pollution level (Fig. 2a). Thereafter, two*

*significant PM$_{2.5}$ concentration decreases occurred around noon and midnight on that day. During the day on 24 December, the mean PM$_{2.5}$ concentration decreased to 73 μg m$^{-3}$ at 07:00 LST. At 11:00 LST, the PM$_{2.5}$ concentration at YZ sharply increased by 221 μg m$^{-3}$ in one hour. At 13:00 LST, the PM$_{2.5}$ concentration at CP decreased from 112 μg m$^{-3}$ to 32μg m$^{-3}$. The PM$_{2.5}$ concentration of AOT stayed below 80 μg m$^{-3}$ until 22:00 LST when it sharply increased by 268 μg m$^{-3}$ in one hour. The following day, the mean PM$_{2.5}$ concentration exceeded 500 μg m$^{-3}$ at 14:00 LST."*

In the discussion of satellite images, it is unclear how you distinguish haze from fog or clouds in the satellite images?

Thank you for the comment. We have rewritten the first sentence in the first paragraph of section 3.2. The sentences now read:

*"The visible channel true color images from the Himawari satellite clearly showed the movement and evolution of the HF. Normally, on the true color satellite images, clouds look white and gray and tend to have texture; haze is usually featureless and pale gray or a dingy white; fog looks similar to the color of clouds but without texture. However, clouds, fog, and haze are sometimes difficult to distinguish from satellite imagery. Hence, we referred to weather phenomena, visibility and PM$_{2.5}$ concentration observed by surface meteorological and air quality stations to distinguish them. A dense fog covered northeastern Tianjin and half of Xianghe county of Heibei Province at 08:00 LST (Fig. 4a)."*

Some of the descriptions used present tense while others used past tense. Be consistent.

Thank you. We have revised present tense into past tense.

The discussion about the HF and foehn characteristics is exhaustion to read. Better rewriting is necessary to improve readability.

Thank you for the comment. We have rewritten the discussion about the HF and foehn characteristics.

Figures are generally in good quality, but figure captions could use more details. For example, the caption for Figure 1 should include a description of the different symbols, although they are described in the text. Also include AWS and PM stations in Figure 1.

Thank you for the comment. We have revised Figure 1 as you suggested.

The font size for the axis labels in the time series plots and some of text in the figures should be enlarged. They are currently too small to read unless the figures are enlarged by 200% (e.g., Fig. 10 c vertical axis, Fig. 10 b, the label for the color bar; Figure 2b).

Thank you for the comment. We have enlarged axis labels, color bar labels and some text in Figure 10 and Figure 2.

Clearly mark the time of HF passage on the time series plots.

Thank you for the comment. We have marked the time of HF passage on the time series plots.

References for review replies:

Li, X., Xia, X., Wang, L., Cai, R., Zhao, L., Feng, Z., Ren, Q., Zhao, K.: The role of foehn in the formation of heavy air pollution events in Urumqi, China. J. Geophys. Res. Atmos., 120, 5371–5384, https://doi.org/10.1002/2014jd022778, 2015.

Vergeiner, J. : South foehn studies and a new foehn classification scheme in theWipp and Inn Valley, PhD thesis, Univ. of Innsbruck, Austria, 2004.

Wang X, Zhang R: Effects of atmospheric circulations on the interannual variation in PM2.5 concentrations over the Beijing–Tianjin–Hebei region in 2013–2018. Atmos Chem Phys 2020, 20(13):7667-7682.